**Data Availability Statement:** All relevant data are within the paper and its Supporting Information files.

# Development of an optimized method for processing peripheral blood mononuclear cells for 1H-nuclear magnetic resonance-based metabolomic profiling

León Gabriel Gómez-Archila[1]ⓔ*, Martina Palomino-Schätzlein[2]ⓔ*, Wildeman Zapata-Builes[3,4], Elkin Galeano[1]ⓔ

**1** Grupo de Investigación en Sustancias Bioactivas, Facultad de Ciencias Farmacéuticas y Alimentarias, Universidad de Antioquia (UdeA), Medellín, Colombia, **2** Servicio de RMN, Centro de Investigación Príncipe Felipe, Valencia, Spain, **3** Grupo Inmunovirología, Facultad de Medicina, Universidad de Antioquia (UdeA), Medellín, Colombia, **4** Grupo Infettare, Facultad de Medicina, Universidad Cooperativa de Colombia, Medellín, Colombia

ⓔ These authors contributed equally to this work.

\* lgabriel.gomez@udea.edu.co (LGGA); mpalomino@cipf.es (MPS)

## Abstract

Human peripheral blood mononuclear cells (PBMCs) are part of the innate and adaptive immune system, and form a critical interface between both systems. Studying the metabolic profile of PBMC could provide valuable information about the response to pathogens, toxins or cancer, the detection of drug toxicity, in drug discovery and cell replacement therapy. The primary purpose of this study was to develop an improved processing method for PBMCs metabolomic profiling with nuclear magnetic resonance (NMR) spectroscopy. To this end, an experimental design was applied to develop an alternative method to process PBMCs at low concentrations. The design included the isolation of PBMCs from the whole blood of four different volunteers, of whom 27 cell samples were processed by two different techniques for quenching and extraction of metabolites: a traditional one using organic solvents and an alternative one employing a high-intensity ultrasound probe, the latter with a variation that includes the use of deproteinizing filters. Finally, all the samples were characterized by 1H-NMR and the metabolomic profiles were compared by the method. As a result, two new methods for PBMCs processing, called Ultrasound Method (UM) and Ultrasound and Ultrafiltration Method (UUM), are described and compared to the Folch Method (FM), which is the standard protocol for extracting metabolites from cell samples. We found that UM and UUM were superior to FM in terms of sensitivity, processing time, spectrum quality, amount of identifiable, quantifiable metabolites and reproducibility.

## Introduction

A peripheral blood mononuclear cell (PBMC) is any blood cell with a round nucleus, such as lymphocyte, monocyte or macrophage [1]. PBMC are composed of three types of cells:

**Funding:** The authors acknowledge the Comité para el Desarrollo de la Investigación (CODI) of the University of Antioquia https://www.udea.edu.co for the financial support of this work. The funder had no role in study design, data collection and analysis, decision to publish, or preparation of the manuscript.

**Competing interests:** The authors have declared that no competing interests exist.

lymphocytes, dendritic cells and monocytes. The abundance or frequency of each type has a characteristic distribution in humans as follows: most PBMC correspond to lymphocytes with an abundance of between 70% and 90%, while dendritic cells are rare with only between 1% and 2% of the total population, and monocytes are in the middle with an abundance ranging from 10% to 20%. There are different types of cells in the lymphocyte family, including $CD3^+$ T cells (70–85%), B-cells (5–10%), and NK cells (5–20%). Specifically, $CD3^+$ lymphocytes are composed of $CD4^+$ and $CD8^+$ T cells with an approximate 2:1 ratio. Furthermore, the activation process of $CD4^+$ T-cells causes a conversion phenomenon in various subsets of effector cells, including Th1, Th2, Th17, Th9, Th22 cells, follicular helpers (Tfh) and different types of regulatory cells [2–5].

In the immune system, PBMCs stand out as an essential component as they are responsible for generating a response to the external agents that enter the body [6, 7]. and to the cells that have transformed into a cancerous cell type [8, 9]. This has led to medical and research interest being shown in studying PBMC in diverse areas such as immunology [10–13], toxicology [14, 15], infectious disease [16–18], allergic diseases [19, 20], cardiovascular diseases [21, 22], hematological malignancies [23], transplant therapy [24], vaccine development [25, 26] and personalized medicine [27]. It is possible to find fundamental information on the function of different types of cells [28], to identify biomarkers and metabolic pathways associated with different medical conditions [29], and to perform disease modeling [30] through *in vitro* studies of PBMCs.

Metabolomic profiling can gain insight into changes in the metabolic state of cells due to external agents, such as diseases and their treatment with medications, environmental factors, diet, lifestyles, genetic effects, toxic exposure, etc. [31–35]. For this purpose, nuclear magnetic resonance (NMR) spectroscopy provides the unbiased detection of metabolites, robustness, reproducibility, minimal sample preparation, easy interpretation and analysis of spectral data, and quantitative information about the metabolome of cells [36, 37].

Previous protocols for the metabolomics profiling of PBMCs by NMR were performed, starting from 20 mL blood and 20 million PBMCs [38]. However, the blood volume available for extracting PBMCs in clinical studies is limited, and the amount of PBMCs in the blood depends on the patient's condition. For instance, immunosuppressive diseases such as HIV-1 infection [39], cancer [40], and primary immunodeficiencies, characterized by low PBMC counts. Therefore, it would be optimal to work with smaller blood and cell samples by optimizing different metabolite extraction method steps, including the cell disruption method, the amount and type of solvents, and the processing time [41–45].

Accordingly, this study aimed to develop an improved method to process PBMCs for metabolomic profiling by NMR spectroscopy. We wished to obtain reproducible and robust high-quality data from PBMC samples of 12.5 million cells or fewer, with minimal sample handling and a short processing time. Our approach is based on using high-intensity ultrasound, as previously described for other types of cells and tissues [46–49], and to compare it to the Folch method, the international standard for processing these biological samples [50, 51]. The method developed for analyzing the metabolic profile of PBMCs by $^1$H-NMR could be a tool for the biomarker identifications associated with disease diagnosis, evaluations of the effects of new treatments on patients, among other approaches [34, 35, 52].

## Materials and methods

### Chemicals and materials

Solvents and reagents were Ficoll-Paque Plus, Phosphate-Buffered Saline (PBS), sodium phosphate dibasic dehydrate and sodium azide, methanol, chloroform, deuterated water and

deuterated trimethylsilyl propanoic acid (TSP-D4) were supplied by Merck (Germany). The ultrapure water was obtained in a Milli-Q purification system of Merck Millipore. The Vivaspin® 500 3,000K MWCO Centrifugal Concentrators were provided by Sartorius.

In order to develop the high-intensity ultrasound process, an LSP-500 ultrasonic liquid processor (Sonomechanics, New York, USA), equipped with an ultrasonic generator of 500 W, an air-cooled piezoelectric transducer (ACT-500), a full-wave Barbell Horn™ (FBH, 21-mm tip diameter) and a reactor chamber (jacketed beaker refrigerated), was used for all the experimental runs.

## Human subjects

This study was approved by the Bioethical Committee, Universidad de Antioquia; and all the individuals signed informed consent, prepared according to Colombian Legislation, Resolution 008430/1993. Healthy controls were 59, 34, 25, and 40 year-old male individuals, recruited through the blood bank from the IPS Universitaria, Universidad de Antioquia. These individuals fulfilled the donor eligibility requirements; their hemoglobin levels equaled or exceeded 13g/dL, or the hematocrit value equaled or was over 39 percent; they were negative for several infectious diseases and the blood samples donation protocol did not require eitherfasting or a nonfasting status. For whole blood, a donation of 450 mL (60-70ml of Citrate Phosphate Dextrose anticoagulant) was obtained from each individual. Four volunteers were analyzed. From the first volunteer, a total of nine samples were processed, a triplicate for each evaluated method (FM, UM, UUM). For the three remaining volunteers, a duplicate of each volunteer was analyzed by each method on two different evaluation days; that is, a total of 27 measurements were made, nine for each evaluated method.

## Human peripheral blood mononuclear cells isolation

The isolation of PBMCs leukocytes was carried out by a Ficoll-Paque gradient method [53]. Peripheral blood, freshly extracted from healthy volunteers, was carefully poured into a tube with Ficoll at the blood/Ficoll 1:4 proportion, and allowed to stand for approximately 20 min, centrifuged at 591 g for 30 minutes at room temperature with brake off to ensure that deceleration did not disrupt the density gradient. Three phases were obtained. At the bottom of the tube, red blood cells (RBCs) and granulocytes concentrated. Likewise, a white fraction of the mononuclear in the middle (PBMCs) and an upper phase corresponding to plasma and platelets formed. The PBMCs phase was carefully transferred to a separate tube and washed with PBS 1X at the 1:10 PBMCs/PBS 1X proportion. Subsequently, PBMC were centrifuged at 1000 g for 5 min at 4°C.

The supernatant was discarded, and the pellet containing PBMCs was resuspended in 1 mL of PBS 1X for cell counting and viability tests. Cells were diluted within a range between 250000 cells/mL and 500000 cells/mL to be counted with the Neubauer chamber. Later an aliquot (50 μL) of the cell suspension was diluted 1:1 (v/v) with 0.4% trypan blue dye (Trypan Blue should be sterile-filtered before using it to do away with the particles in solution that would disturb the counting process). After carefully the hemocytometer chamber was filled with 20 μL of cell suspension. Then it was incubated for 1–2 min at room temperature (incubations exceeding 30 min may cause decreased cell viability due to Trypan toxicity). Nonviable cells are blue and viable cells are unstained. Next, viable cells were counted under the microscope in four 1 x 1 mm squares of the Neubauer chamber and the average number of cells per square was determined (Neubauer hemocytometer consists of two chambers, each divided into 9 $mm^2$ squares) [54–57]. Finally, cells were diluted in PBS 1X to obtain individual portions of 12.5 million viable cells in 100 μL. PBMC were frozen at -80°C until processed.

## Extraction of metabolites for the $^1$H-NMR experiments

Frozen samples were placed on ice for 5 min, and then subjected to the first extraction procedures, the Folch Method (FM) [58, 59]. Briefly, 160 μL of methanol and 80 μL of chloroform at 4˚C were added per tube (12.5 million cells). Samples were then homogenized by vortexing and left to stand for 15 min. For uniform cell breakage, samples were submitted to three freeze-thaw cycles with liquid nitrogen. Then 125 μL of distilled water and 125 μL of chloroform were added to each sample, which were later vortexed. Samples were then centrifuged at 15000 g for 30 min at 4˚C to separate phases. The solution was separated into an upper water/methanol phase (with polar metabolites, aqueous phase), an interphase containing mainly proteins, DNA/RNA, cell membranes, and a lower chloroform/methanol phase (with lipophilic compounds, organic phase). The aqueous phase was lyophilized overnight to obtain dry extracts. Extracts were stored at -80˚C until sample preparation for the $^1$H-NMR experiments.

The second procedure was the Ultrasound Method (UM), in which 650 μL of phosphate buffer (50 mM $Na_2HPO_4$ pH 7.4, in $D_2O$ with 0.1 mM of deuterated trimethylsilyl propanoic acid (TSP-D4)) were added per sample with 12.5 million cells. Cells were resuspended with a micropipette and homogenized with a vortex. Then for cell breakage, samples were submitted to a cycle described as follows: First cells were frozen with liquid nitrogen (1 min). Then the sample was immersed in a refrigerated bath (4˚C) equipped with a high-intensity ultrasound probe set at a frequency of 20050 Hz and a 100% amplitude for 5 min. This cycle was run six times. Samples were then centrifuged at 12000 g for 120 min at 4˚C to separate phases. The solution was separated into an upper phase (with metabolites) and a lower phase (containing mainly proteins, DNA/RNA, and cell membranes). The supernatant was stored at -80˚C until the sample analysis for the $^1$H-NMR experiments.

The third procedure was called the Ultrasound and Ultrafiltration Method (UUM). It involves taking the supernatant obtained from the second method and filtered with a ultrafilter previously washed with phosphate buffer (10X with phosphate buffer, pH 7.4). Samples were then centrifuged at 12000 g for 120 min at 4˚C and the filtered solution was stored at -80˚C until the sample analysis for the $^1$H-NMR experiments.

## $^1$H-NMR experiments

The freeze-dried powder from FM was solubilized in 650 μL of phosphate buffer (50 mM $Na_2HPO_4$, pH 7.4, in D2O) containing 0.1 mM of deuterated trimethylsilyl propanoic acid (TSP-D4) as a reference standard and 550 μL were transferred to a 5 mm NMR tube. Likewise, 550 μL of supernatant, or filtered from UM and UUM, were transferred to independent 5 mm NMR tubes. All the samples (FM, UM, or UUM method) were stored at 4˚C, equilibrated at room temperature for 15 min before analyzing, which took place on the same day. The $^1$H-NMR spectra of extracts were recorded at 300K by a Bruker AVANCE III 600.13 MHz spectrometer, equipped with 5 mm triple-resonance z-gradient cryoprobe (Prodigy® TCI, 1H-13C/15N-2H). TopSpin, version 3.6.2 (Bruker GmbH, Karlsruhe, Germany), was used for spectrometer control purposes. $^1$H 1D Nuclear Overhauser Effect Spectroscopy (NOESY) NMR spectra, with water presaturation and spoil gradients (*noesygppr1d* pulse sequence), were acquired with 256 free induction decays (FIDs), 64k data points, a spectral width of 30 ppm, and a relaxation delay of 60 s. Total Correlation Spectroscopy (TOCSY) and multiplicity Heteronuclear Single Quantum Correlation (HSQC) were performed on representative samples with 256–512 t1 increments, 32–96 transients and a relaxation delay of 1.5 s. The TOCSY spectra were recorded by a standard MLEV-17 pulse sequence with mixing times (spin-lock) of 65 ms.

## Data analysis and statistics

**NMR spectra processing.** [1]H-NMR spectra were transformed with a 0.5 line-broadening, and manually baseline- and phase-corrected with Topspin 4.0.9. NMR signals of TSP-D4 were referenced to 0 ppm. For metabolite identification purposes, the [1]H and chemical shift values and multiplicity of signals were compared with the reference data from the Chenomx software (Chenomx NMR Suite 8.4, Chenomx Inc., Edmonton, Canada) in combination with spectral databases Human Metabolome Database and the Biological Magnetic Resonance Bank and several literature reports [33, 60]. Optimal integration regions were defined for each metabolite in an attempt to select signals without overlapping. Integration was performed with Mestre-Nova 14 (Mestrelab Research, SL, Santiago de Compostela, Spain) by performing a manual integration of the previously identified signals. With these regions, an integration matrix (Integral Regions) was built, which was later applied to the 27 acquired spectra and a matrix of integrals was built for all the spectra (Integral series). This matrix of integrals was normalized by the sum of the total signals of the spectrum using Excel® (Microsoft, USA).

**Quantification and comparison of metabolic profiles.** An analysis of the areas of every variable (metabolite) was carried out for each method (FM, UM and UUM) in the 27 spectra analyzed as follows: measures of central tendency (mean or median) were determined with Excel®, and statistical assumptions of normality (Shapiro-Wilk Normality Test) and homoskedasticity (Levene test) were evaluated with RStudio. To determine the difference in the measures of central tendency between the methods, a comparison of the gold standard for the processing of biological samples (FM) with the developed methods (UM and UUM) was made. Resulting in two comparisons, FM versus UM and FM versus UUM. The difference in measures of central tendency was determined as appropriate; mean difference using t-Test for paired samples (if the variable in both methods had normal distribution and homoskedasticity) or median difference using Wilcoxon Test for k paired samples (if the variable in one of the methods had non normal distribution and/or heteroskedasticity). For tests with a p value less than 0.05 ($p < 0.05$), a statistically significant difference between the means (mean or median as appropriate) is assumed for the variable evaluated. This last statistical test was performed with the RStudio software.

**Analysis of repeatability and reproducibility.** The *Six Sigma Gage R&R Measure* (Part of the Six Sigma package of the R software) was applied to evaluate repeatability and reproducibility of three methods. The analysis was performed with the three patient samples analyzed on different days and by different analysts. A total of 18 samples were analyzed, 1 sample per patient, per day and per method, in 2 evaluation days. The input data were the variables that presented statistically significant changes ($p < 0.05$) in the central (measures of central tendency) difference test (t-Test or Wilcoxon Test as applicable). The coefficient of variation (CV) per variable of each method was compared. The repeatability of the methods was analyzed by determining the CV between replicates of the same patient, analyzed by NMR on the same day and processed by the same analyst. On the other hand, for the reproducibility analysis, the CV was determined between samples from the same patient, analyzed by NMR but processed in two days and by different analysts. Results were represented graphically.

**Determination of limit of detection and limit of quantitation.** Three regions were selected in the NMR spectrum (0.5 ppm, 6.5 ppm, 9.5 ppm) and had the lowest possible noise or interference level. The integration process of these regions was carried out (in the same way as with the other metabolites in the samples) in the 27 spectra analyzed, and the limit of detection (LOD) and the limit of quantification (LOQ) of all three methods (FM, UM and UUM) were determined. All the statistical analyses were performed with the RStudio and Excel software.

## Results

### Optimization of the pretreatment of samples and metabolite extraction

Equal amounts of isolated PBMCs were processed by the new protocols (Fig 1) and the results were compared to the Folch method (FM) [50, 51], which is widely used in mammalian cell metabolomics and has been previously applied to PBMCs [38]. In the new methods, cells were mixed with an aqueous buffer, and not with toxic solvents, such as methanol and chloroform in FM. Then cycles of freezing and cell disruption with a high-intensity sonicator were applied to the cell suspension, and the resulting supernatant was transferred directly to an NMR tube to be analyzed (in the UM method).

The cell freezing and disruption processes were optimized as follows: starting with a simple cycle to immerse the sample into liquid nitrogen (1 min), combined with high-intensity ultrasound at 20,050 Hz, and 50% amplitude for 5 min (in a cold bath at 4˚C). However, under these conditions, the necessary cell disruption was not achieved as verified by visual inspection and a microscope. Therefore, the parameters of the ultrasound equipment were increased from 50% to 100% of the amplitude for the same period time. The number of cycles (freezing + cell disruption by ultrasound) was increased one by one until six cycles, which was the amount required to achieve a successful quenching and extracting process with the sample. For the UM centrifugation process, we based our work on previous cell disruption and metabolite isolation protocols [46, 47]. However, we had to apply more time and more power during centrifugation to eliminate membranes and cell debris as the volume of PBMC cells is much smaller than most other cells (e.g., HeLa cells have an average volume of 3000 $\mu m^3$, while the volume of PBMCs is only 130 $\mu m^3$). We determined that a cycle of 12000 g for 120 min at 4˚C would be required to separate the pellet from the solution. An increase in centrifugation force or time did not improve the subsequent NMR results, and shorter centrifugation times did not achieve optimal cell pellet separation. When analyzing the UM $^1$H-NMR spectra results, we

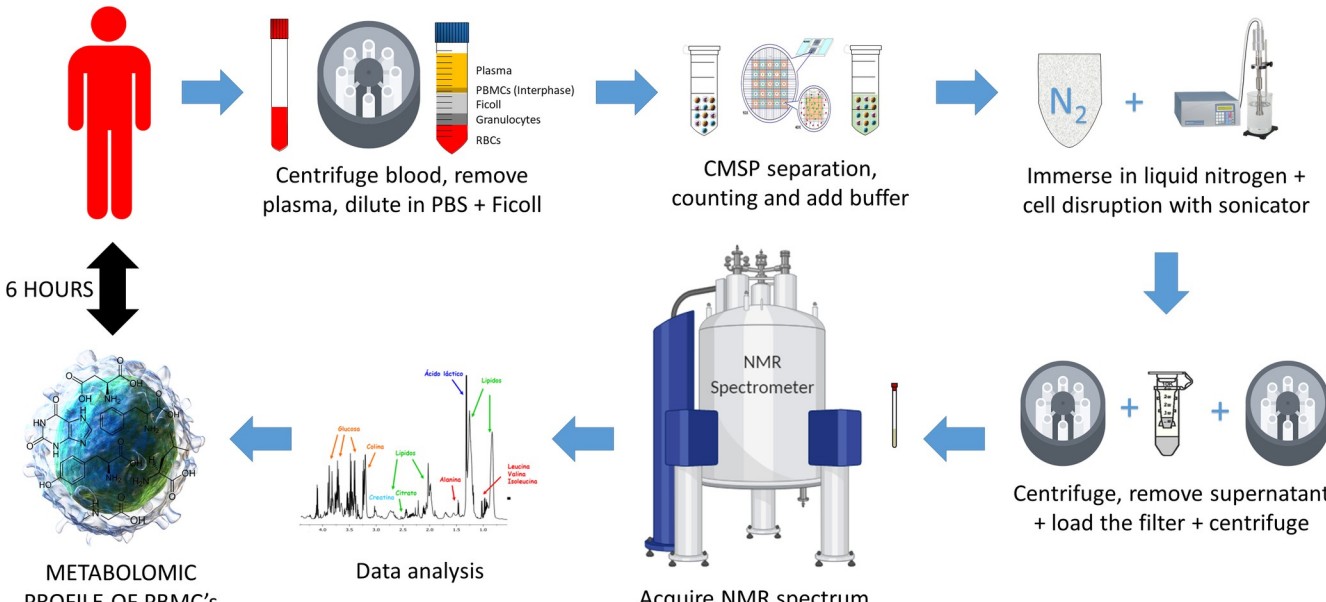

**Fig 1. Step by step UUM method.** Schematic explanation of PBMC processing by the UUM method (Ultrasound and Ultrafiltration Method). Using these methods, it is possible to obtain the metabolomic profile in 6 h starting from whole blood. The PBMC samples were isolated from the peripheral blood of healthy human individuals. Samples were split into an aliquot (12.5 million cells) for characterization. Finally, PBMC were extracted and the $^1$H-NMR metabolic profiles were determined.

detected the presence of proteins in the spectrum, which is a generally undesirable characteristic for metabolomic profiling. Therefore, as a second option, the suspension was first transferred to a centrifugal filter to eliminate proteins, re-centrifuged and then moved to an NMR tube (for the UUM method). In both cases (methods UM and UUM), an NMR spectrum was obtained for PBMCs in less than 6 h, while FM required several hours for solvent elimination by freeze-drying and evaporation.

## Metabolic profile of PBMCs

The representative $^1$H-NMR spectra resulting from the three tested methods are shown in Fig 2. An assignment of the different signals in the spectra was performed with the help of 2D-NMR spectra, and with information available from public databases, the software Chenomx® 8.6 (Alberta, Canada) and existing literature about the metabolic content of PBMCs [38]. As a result, it was possible to identify more than 40 different metabolites (Fig 3). The primary metabolites were organic acids, amino acids and nucleotides.

A comparison of the metabolites that could be quantified and identified in the spectra resulting from each extraction method is found in Table 1. For FM, 37 metabolites, consisting in 17 organic acids, 18 amino acids and two nucleotides, were identified. For UM and UUM, 43 metabolites consisting in 19 organic acids, 19 amino acids and five nucleotides, were identified.

It should also be noted that not all the metabolites identified in spectra had an optimal quantification quality for the later metabolomics analysis. In Table 1, the quality of the signals from each metabolite in all the methods is classified (present, increased presence, absent, unquantifiable). The most noteworthy case was FM, for which 37 metabolites were identified, but only 31 were quantifiable (Table 1).

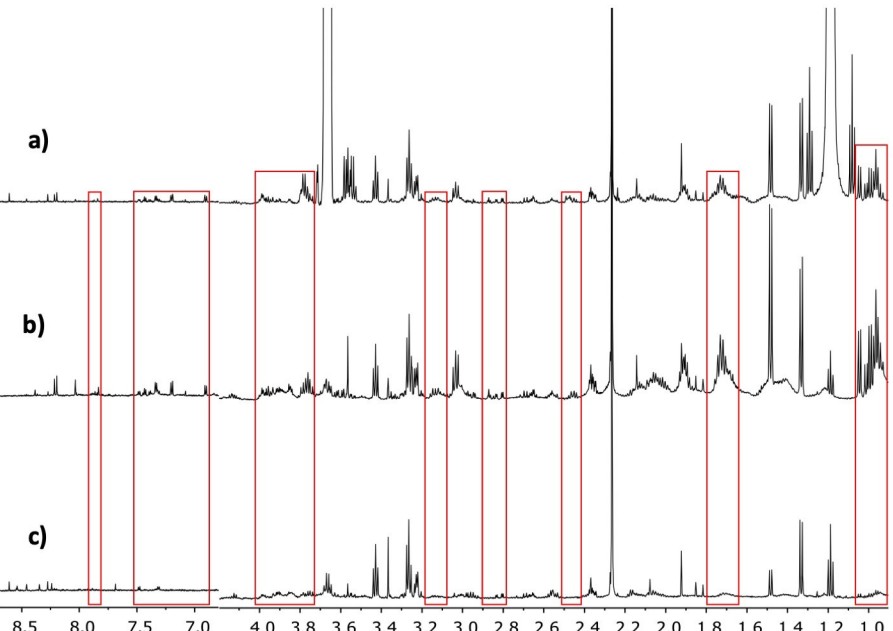

**Fig 2. Comparison of the spectra obtained from the three different PBMC processing methods evaluated for metabolomic profiling.** a) Ultrasound and Ultrafiltration Method, b) Ultrasound Method, c) Folch Method. The differences in specter regions between methods are seen in red boxes.

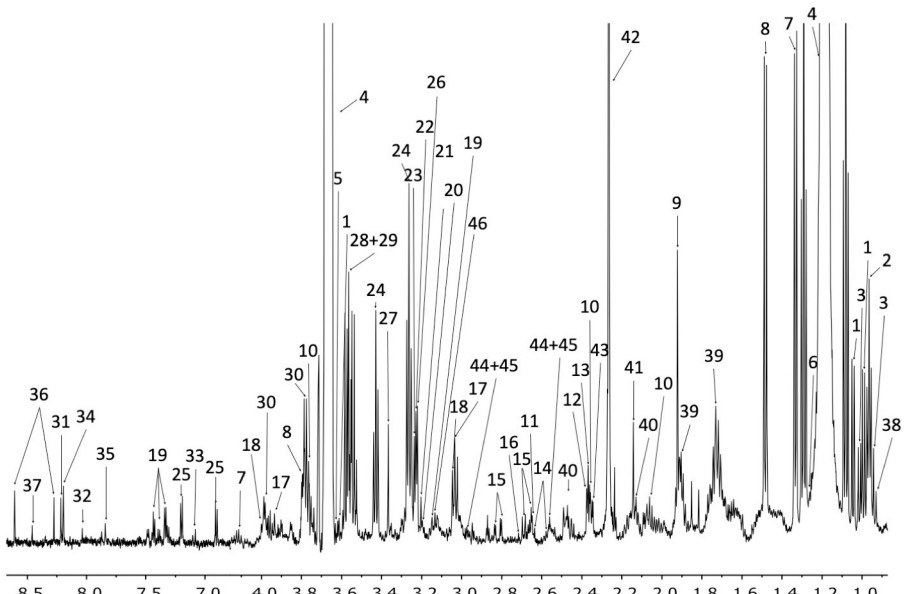

**Fig 3. The assigned UUM $^1$H-NMR spectrum of PBMCs.** Metabolite assignments are indicated by the numbers 1. Valine, 2. Leucine, 3. Isoleucine, 4. Ethanol, 5. Threonine, 6. 3-Hydroxyisovalerate, 7. Lactate, 8. Alanine, 9. Acetate, 10. Glutamate, 11. Methionine, 12. Pyruvate, 13. Succinate, 14. Citrate, 15. Aspartate, 16. Sarcosine, 17. Creatine, 18. Creatinine, 19. Phenylalanine, 20. Choline, 21. O-Phosphocoline, 22. Carnitine, 23. Betaine, 24. Taurine, 25. Tyrosine, 26. Trimethylamine N-oxide, 27. Methanol, 28. Glycine, 29. Glycerol, 30. Serine, 31. Inosine, 32. GTP, 33. Xanthurenate, 34. Oxypurinol, 35. Xanthine, 36. AMP, 37. Formate, 38. 2-hydroxybutyrate, 39. Lysine, 40. Glutamine, 41. Hydroxyacetone, 42. Acetoacetate, 43. Methylacetoacetate, 44. Reduced glutathione (GSH), 45. Oxidized glutathione (GSSG), 46. Malonate, 47. Trimethylamine.

## Quantitative comparison of the PBMCs metabolomic profiles

The normalized concentration values of the all metabolites (S1 and S2 Tables) were compared by applying a Student's t-Test (normal distribution and homoscedasticity) or a Wilcoxon Test (non normal distribution and/or heteroskedasticity). From this analysis, the number of metabolites lowered to 20, for which statistically significant differences were found for both comparisons. The results presented in Table 2 show the differences obtained between the means of the two established comparisons FM vs. UM and FM vs. UUM.

For a more detailed analysis, eight representative metabolites with isolated well-defined signals in all the spectra were selected (Valine, Alanine, Hydroxyacetone, Creatine, Creatinine, Choline, Taurine, Inosine). A repeatability and reproducibility analysis of each method was carried out by the Six Sigma Gage R&R Measure command. The results of this analysis are shown in Fig 4 and S1–S8 Figs.

In Fig 4, the graphs on the left represent the range control chart evaluated per metabolite with their corresponding control limits adapted for R&R studies. In a method or process with good repeatability, all the points should lie within the control limits. For the FM method, a point (Taurine concentration range of patient 3) is observed outside the upper control limit. In addition, FM has ranges and, therefore, the dispersion of the values between the replicates for the same patient are wider than those presented in methods UM and UUM. For this reason, we can affirm that UM and UUM are methods with greater repeatability than FM.

The graphs on the right show all the measurement points (Concentration of each metabolite) by method (x-axis). The tracer line represents the mean of each one. These graphs allow an analysis of the methods' reproducibility as the dispersion of the data between patients can be observed on different days for each method. The graphs show how data dispersion

**Table 1. List and detail of the metabolites assigned in the different methods evaluated.**

| METABOLITE | $\delta$ in ppm (Multiplicity) | CT | GA (Visual) | | | FO (%) | | |
|---|---|---|---|---|---|---|---|---|
| | | | FM | UM | UUM | FM | UM | UUM |
| 2-hydroxybutyrate | 0,89 (t), 1.61 (m), 1.65 (m), | OA | + | + | + | 100 | 100 | 100 |
| 3-Hydroxyisovalerate | 1.25 (s) | OA | + | + | + | 100 | 100 | 100 |
| Acetate | 1.92 (s) | OA | + | + | + | 100 | 100 | 100 |
| Acetoacetate | 2.26 (s) | OA | + | + | + | 100 | 100 | 100 |
| Alanine | 1.48 (d), 3.80 (q) | AA | + | + + | + + | 100 | 100 | 100 |
| AMP | 8.27 (s), 8.61 (s) | N | + | + | + | 100 | 100 | 100 |
| Aspartate | 2.67 (m), 2.81 (m) | OA | + | + | + | 100 | 100 | 100 |
| Betaine | 3.24 (s), 3.89 (s) | AA | + | + + | + + | 100 | 100 | 100 |
| Carnitine | 3.23 (s) | AA | + | + | + | 100 | 100 | 100 |
| Choline | 3.20 (s), 4.04 (m) | OA | + | + | + | 100 | 100 | 100 |
| Citrate | 2.56 (d), 2.63 (d) | OA | + | + | + | 100 | 100 | 100 |
| Creatine | 3.03 (s), 3.92 (s) | AA | + | + + | + | 100 | 100 | 100 |
| Creatinine | 3.05 (s), 4.01 (s) | AA | + | + + | + + | 89 | 100 | 100 |
| Formate | 8.46 (s) | OA | - | + | + | 33 | 100 | 100 |
| Glutamate | 2.05 (m), 2.36 (m), 3.76 (m) | OA | + | + | + | 100 | 100 | 100 |
| Glutamine | 2.12 (m), 2.48 (m) | AA | - | + | + | 44 | 100 | 100 |
| Glycine | 3.57 (s) | AA | + | + + | + + | 100 | 100 | 100 |
| GSH+GSSG | 2.53–2.58 (m), 2.96 (dd) | AA | + | + | + | 100 | 100 | 100 |
| GTP | 8.03 (s) | N | O | + + | + | 33 | 100 | 89 |
| Hydroxyacetone | 2.14 (s) | OA | + | + + | + + | 100 | 100 | 100 |
| Inosine | 6.14 (d), 8.19 (s), 8.22 (s) | N | + | + | + | 78 | 100 | 100 |
| Isoleucine | 0.94 (t), 1.01 (d) | AA | + | + | + | 100 | 100 | 100 |
| Lactate | 1.33 (d), 4.11 (q), 4.12 (q), 4.13 (q) | OA | + | + | + | 100 | 100 | 100 |
| Leucine | 0.96 (t) | AA | + | + + | + + | 100 | 100 | 100 |
| Lysine | 1.72 (m), 1.91 (m) | AA | + | + + | + + | 100 | 100 | 100 |
| Malonate | 3.13 (s) | OA | - | + | + | 89 | 100 | 100 |
| Methionine | 2.17 (m), 2.65 (t), 3.85 (dd) | AA | + | + | + | 100 | 100 | 100 |
| Methylacetoacetate | 2.34 (s) | OA | + | + | + | 100 | 100 | 100 |
| O-Phosphocoline | 3.21 (s) | AA | + | + | + | 100 | 100 | 100 |
| Oxypurinol | 8.38 (s) | N | O | + | + | 33 | 100 | 78 |
| Phenylalanine | 3.14 (m), 3.30 (m), 7.34 (d), 7.38 (d), 7.43 (t) | AA | - | + | + | 44 | 100 | 100 |
| Pyruvate | 2.38 (s) | OA | + | + | + | 100 | 100 | 100 |
| Sarcosine | 2.72 (s) | AA | - | + | - | 56 | 100 | 44 |
| Serine | 3.79 (dd), 3.99 (m) | AA | - | + | + | 100 | 100 | 100 |
| Succinate | 2.37 (s) | OA | + | + | + | 100 | 100 | 100 |
| Taurine | 3.26 (t), 3.43 (t) | OA | + | + | + | 100 | 100 | 100 |
| Treonine | 3.59 (d), 4.22 (m) | AA | + | + | + | 78 | 100 | 56 |
| Trimethylamine | 2.87 (s) | OA | O | + + | + | 56 | 100 | 100 |
| Trimethylamine N-oxide | 3.22 (s) | OA | + | + | + | 100 | 100 | 100 |
| Tyrosine | 3.96 (dd), 6.90 (d), 7.20 (d) | AA | O | + + | + | 22 | 100 | 100 |
| Valine | 0.99 (d), 1.04 (d), 2.29 (m), 3.62 (d) | AA | + | + + | + + | 100 | 100 | 100 |
| Xanthine | 7.83 (s) | N | O | + + | + | 33 | 100 | 100 |
| Xanthurenate | 7.08 (dd) | OA | O | + + | + | 33 | 100 | 100 |

FM: Folch Method, UM: Ultrasound Method, UUM: Ultrasound and Ultrafiltration Method, $\delta$: chemical shift, (s): singlet, (d): duplet, (dd): double doublet, (t): triplet; (q): quartet, (m): multiplet, CT: Compound Type, OA: Organic Acid, AA: Amino Acid, N: Nucleotide, GA: Graphical Analysis, +: Present, ++: Increased presence O: Absent, -: Unquantifiable, FO: frequency of occurrence. For FO less than or equal to 33%, the individual integration values were evaluated to define presence or absence. If the integration mean was less than 0, it was determined as absent in the method samples.

**Table 2. Metabolites significant for the comparisons between FM vs. UM and FM vs. UUM.**

| Metabolite | FM | | UM | | UUM | | FM | UM | UUM | LT | FM vs UM | FM vs UUM | Variation | |
|---|---|---|---|---|---|---|---|---|---|---|---|---|---|---|
| | Mean | Med | Mean | Med | Mean | Med | S-W p-Value | S-W p-Value | S-W p-Value | p-Value | Mean Comparison | Mean Comparison | FM vs UM | FM vs UUM |
| 2-hydroxybutyrate | 5.50 | 4.76 | 13.88 | 14.85 | 16.15 | 17.87 | 0.615 | 0.105 | 0.113 | 0.066 | 1.75.E-04 | 7.39.E-04 | ↓ | ↓ |
| Alanine | 18.87 | 16.30 | 43.88 | 38.92 | 35.39 | 29.71 | 0.141 | 0.070 | 0.106 | 0.260 | 4.64.E-06 | 6.85.E-04 | ↓ | ↓ |
| AMP | 4.87 | 4.86 | 1.60 | 1.44 | 2.03 | 1.89 | 0.483 | 0.545 | 0.748 | 0.171 | 1.53.E-05 | 5.73.E-05 | ↑ | ↑ |
| Carnitine | 15.18 | 15.88 | 8.44 | 8.39 | 10.98 | 11.15 | 0.085 | 0.848 | 0.120 | 0.099 | 4.06.E-06 | 6.97.E-04 | ↑ | ↑ |
| Choline | 7.12 | 6.01 | 4.67 | 4.96 | 5.83 | 5.14 | 0.225 | 0.207 | 0.283 | 0.161 | 2.32.E-02 | 4.90.E-02 | ↑ | ↑ |
| Citrate | 12.43 | 12.45 | 9.25 | 9.10 | 4.58 | 4.71 | 0.839 | 0.600 | 0.319 | 0.038 | 1.95.E-02 | 3.91.E-03 | ↑ | ↑ |
| Creatine | 2.20 | 1.79 | 9.97 | 10.09 | 6.02 | 7.06 | 0.047 | 0.032 | 0.281 | 0.197 | 3.91.E-03 | 3.91.E-03 | ↓ | ↓ |
| Creatinine | 1.80 | 1.43 | 6.21 | 6.38 | 4.19 | 4.53 | 0.057 | 0.166 | 0.133 | 0.083 | 2.85.E-06 | 3.25.E-05 | ↓ | ↓ |
| Glutamate | 36.55 | 33.61 | 64.27 | 63.90 | 18.37 | 17.75 | 0.119 | 0.065 | 0.804 | 0.318 | 1.38.E-06 | 9.07.E-07 | ↓ | ↑ |
| GSH+GSSG | 6.11 | 6.09 | 10.22 | 10.44 | 1.91 | 2.30 | 0.983 | 0.467 | 0.473 | 0.542 | 2.50.E-03 | 5.40.E-04 | ↓ | ↑ |
| Hydroxyacetone | 5.16 | 5.09 | 8.18 | 8.12 | 6.87 | 6.83 | 0.444 | 0.445 | 0.583 | 0.069 | 4.13.E-08 | 1.33.E-04 | ↓ | ↓ |
| Inosine | 1.07 | 0.94 | 3.10 | 3.17 | 3.42 | 3.33 | 0.590 | 0.120 | 0.481 | 0.029 | 3.91.E-03 | 3.91.E-03 | ↓ | ↓ |
| Isoleucine | 2.88 | 2.52 | 9.10 | 8.44 | 7.07 | 4.52 | 0.282 | 0.014 | 0.006 | 0.145 | 3.91.E-03 | 3.91.E-03 | ↓ | ↓ |
| Leucine | 14.67 | 12.12 | 42.93 | 40.20 | 28.11 | 22.97 | 0.007 | 0.131 | 0.039 | 0.686 | 3.91.E-03 | 3.91.E-03 | ↓ | ↓ |
| Lysine | 32.74 | 27.16 | 94.86 | 90.05 | 60.29 | 55.44 | 0.010 | 0.090 | 0.445 | 0.589 | 3.91.E-03 | 3.91.E-03 | ↓ | ↓ |
| Methionine | 13.39 | 12.55 | 11.04 | 10.56 | 7.70 | 7.61 | 0.745 | 0.006 | 0.183 | 0.198 | 7.81.E-03 | 1.28.E-05 | ↑ | ↑ |
| Methylacetoacetate | 1.19 | 1.00 | 2.61 | 2.67 | 0.73 | 0.66 | 0.050 | 0.328 | 0.149 | 0.445 | 7.81.E-03 | 2.73.E-02 | ↓ | ↑ |
| Taurine | 51.04 | 50.39 | 21.06 | 19.64 | 40.21 | 38.88 | 0.570 | 0.018 | 0.179 | 0.154 | 3.91.E-03 | 1.10.E-03 | ↑ | ↑ |
| Trimethylamine N-oxide | 13.04 | 13.57 | 7.26 | 6.56 | 8.27 | 8.51 | 0.061 | 0.029 | 0.533 | 0.000 | 3.91.E-03 | 3.91.E-03 | ↑ | ↑ |
| Valine | 3.69 | 3.08 | 12.08 | 10.46 | 11.16 | 7.35 | 0.027 | 0.182 | 0.004 | 0.155 | 3.91.E-03 | 3.91.E-03 | ↓ | ↓ |

FM: Folch Method, UM: Ultrasound Method, UUM: Ultrasound and Ultrafiltration, Med: Median, S-W p-Value: Shapiro-Wilk (Normality Test) p-Value, LT: Levene test (Homoskedasticity test), Mean comparison: using t-test or Wilcoxon test, Variation: ↓ (Smaller area or relative concentration in the reference method) ↑ (Bigger area or relative concentration in the reference method).

(scattered points on the y-axis of each inflection point of the tracer line) for FM is generally wider compared to UM and UUM. This was confirmed by comparing the coefficients of variation (CV) for each method, which are observed at the bottom of each graph per evaluated metabolite. Once again, FM has higher CVs for most of the metabolites evaluated against UM and UUM, with an average CV of 45%, compared to 16% and 18% for UM and UUM, respectively. To further complete this numerical comparison, we also made a graphical comparison by overlapping the zoomed signals of the selected metabolites, as depicted in Figs 5 and 6.

Finally, we also calculated the LOD and LOQ of the three methods, as presented in Fig 7. Once again, we obtained better results for UUM and UM than for FM, and slightly better results for UUM than for UM. This result was expected as we were able to identify and quantify more metabolites with methods UUM and UM.

## Discussion

In this work, we present two new methods for determining the metabolic profile of PBMCs by NMR, which were compared with the FM method by evaluating processing and quality parameters. As a quantitative comparison proved (Table 2), significant differences in the normalized metabolite concentration between the new methods (UM and UUM) and the FM method existed. It is worth mentioning that these analyzes were performed with a small sample size,

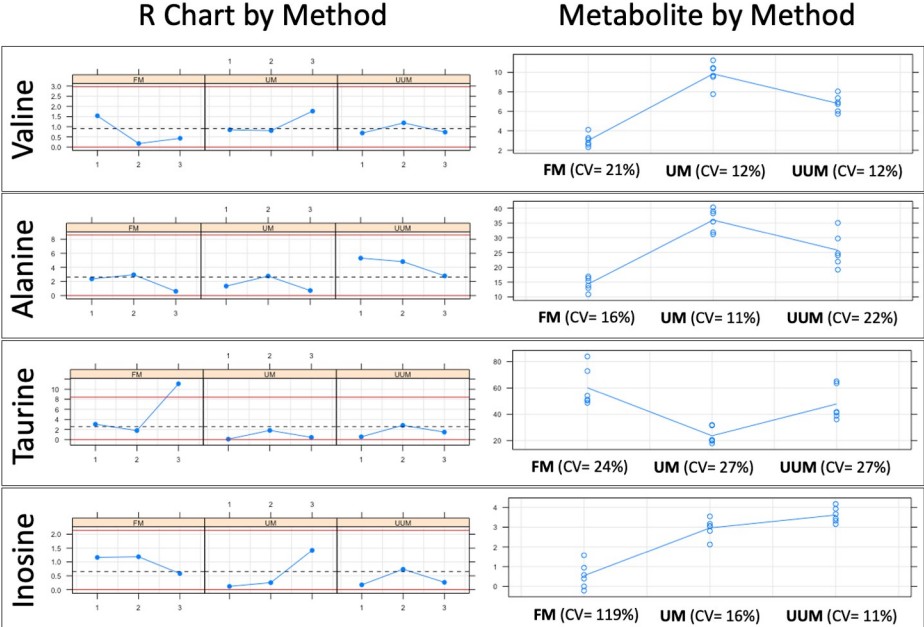

**Fig 4. Six Sigma Gage R&R Measure.** A comparative repeatability and reproducibility analysis (FM vs. UM vs. UUM) for metabolites Valine, Alanine, Taurine and Inosine was carried out using the Six Sigma tool from RStudio. FM: Folch Method, UM: Ultrasound Method, UUM: Ultrasound and Ultrafiltration Method, R Chart by Method: Range chart by method (Numbers 1, 2, and 3 on the x-axis: Patients evaluated; Differences in normalized concentration values on the y-axis), Metabolite by Method: Metabolite concentration of samples by method (Methods evaluated on the x-axis; Normalized concentration values on the y-axis), CV: coefficient of variation as a percent.

which may affect the power of the statistical tests performed. For example, the normality test (Shapiro-Wilk test) was used as a means to select whether to take a parametric or non-parametric approach to testing the hypothesis and contrast the methods and its metabolites. In

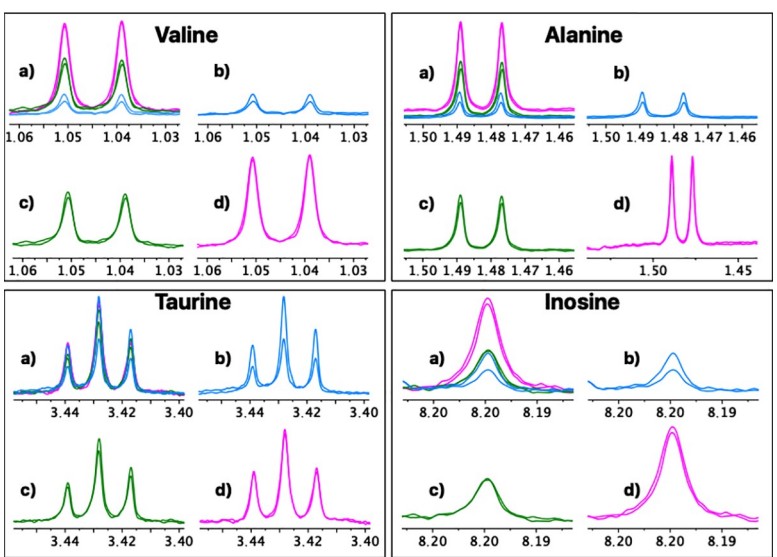

**Fig 5. Comparative repeatability graphical analysis (FM vs. UM vs. UUM) for metabolites Valine, Alanine, Taurine and Inosine.** a) Three overlapping methods, b) Folch method, c) Ultrasound and ultrafiltration method, d) Ultrasound method. A duplicate spectrum, acquired for the same patient and on the same day, is shown for each method.

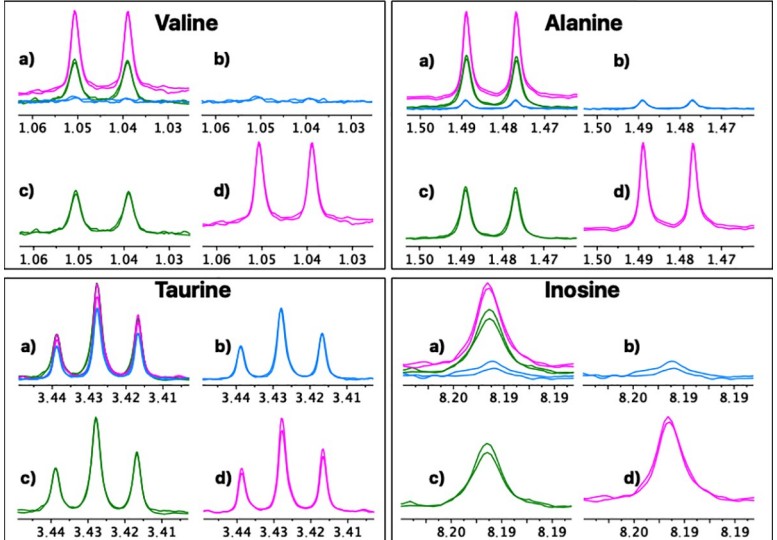

**Fig 6. Comparative reproducibility graphical analysis (FM vs. UM vs. UUM) for metabolites Valine, Alanine, Taurine and Inosine.** a) Three overlapping methods, b) Folch method, c) Ultrasound and ultrafiltration method, d) Ultrasound method. For each method a duplicate spectrum is shown for the same patient but processed on different days.

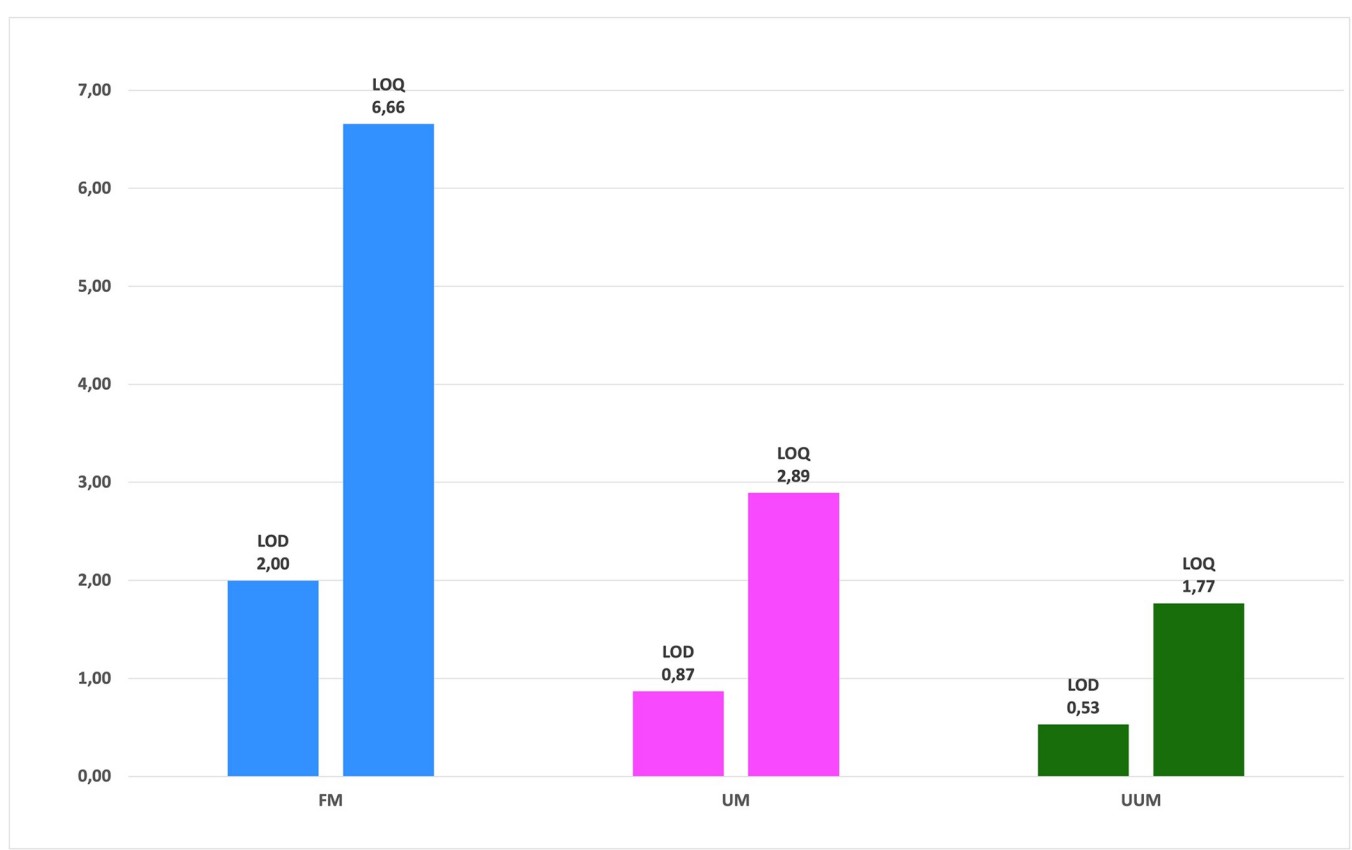

**Fig 7. Sensitivity comparison of the FM, UM and UUM methods with the limit of detection and limit of quantification.** Calculated by the standard deviation (LOD = 3xSD and LOQ = 10xSD) of the integrals of the three selected noise regions of the spectrum (0.5 ppm, 6.5 ppm and 9.5 ppm) for all the spectra of each method. Normalized concentration values on the y-axis.

**Table 3. General comparison of the three methods tested for the metabolomics profiling of PBMCs by NMR.**

| FEATURE | Method | | |
|---|---|---|---|
| | **FM** | **UM** | **UUM** |
| **Number of cells** | 12.5 million cells | 12.5 million cells | 12.5 million cells |
| **Processing time** | 16 hours | 4 hours | 6 hours |
| **Processing steps** | 7 | 4 | 5 |
| **Lyophilization** | Yes | No | No |
| **Presence of Protein** | No | Yes | No |
| **Solvents** | Methanol, $H_2O$, $CHCl_3$ | $H_2O$ | $H_2O$ |
| **Amount of metabolites (detected–quantifiable)** | 37–31 | 43–43 | 43–42 |
| **R&R test[1]** | 36% | 15% | 23% |
| **Repeatability[2]** | 10% | 3% | 3% |
| **Reproducibility[3]** | 30% | 5% | 10% |
| **Limit of detection[4]** | 2.00 | 0.87 | 0.53 |
| **Limit of quantification[4]** | 6.66 | 2.89 | 1.77 |

FM: Folch Method, UM: Ultrasound Method, UUM: Ultrasound and Ultrafiltration Method, +: Acceptable, ++: Good, +++: Very good.

[1]Overall Coefficient of Variation for the R&R test for the eight evaluated metabolites.

[2]Coefficient of variation between replicates of the same patient (processed on the same day) for the concentration of the eight evaluated metabolites (as an average of individual CVs)

[3]Coefficient of variation between replicates of the same patient (processed on two different days) for the concentration of eight evaluated metabolites (as an average of individual CVs), Calculated with the standard deviation (LOD = 3xSD and LOQ = 10xSD) of the integrals of the three selected noise regions of the spectrum (0.5 ppm, 6.5 ppm, 9.5 ppm) for all the spectra from each method.

these case, normality and t test have little power to reject the null hypothesis because small samples most often pass normality tests. But, both tests are designed to achieve reliable results with small sample sizes. (n<30) [61].

In Table 3, we present a comparison of the main parameters of the three methods (FM, UM, UUM). Our results suggest that the best method to analyze PBMCs for NMR metabolomic profiling was UUM for sensitivity, repeatability and reproducibility, but also processing time and robustness (related to the number of steps to be performed).

Of the advantages that the new method offers, it is worth highlighting the fewer PBMCs required to obtain quantifiable results. A previous work has employed 20 million cells [38], which is twice the number of cells herein used, but with similar spectral quality. When working with 12.5 million of PBMCs, we obtained a better LOQ and LOD with the new methods UM and UUM *versu*s FM, which identified more metabolites (43, 43 and 37, respectively), and, most importantly, many more quantifiable metabolites (43, 42 and 31, respectively). This increase in the number of metabolites means more variables to be analyzed in a metabolomic study and, therefore, an increased probability of identifying a biomarker and metabolic pathway associated with a biological process or medical condition, which summarizes the goal of metabolomics. However, it should be noted that the use of centrifugal filters caused two unwanted signals to appear in the spectrum that resulted from the filter's own composition (Glycerol at 3.56 ppm) or the washing solution (Ethanol at 3.67 ppm), as seen in the UUM spectrum in Fig 2. Fortunately, these signals did not overlap the signals of the other metabolites detected in the spectrum from UUM.

Furthermore, FM is very efficient in removing metabolites that could interfere with the analysis and are present in the biological matrix (proteins, lipids, and other interferences). Therefore, it has an advantage over the UM method, where it was still possible to observe wide bands of protein and lipoprotein signals (see the red boxes in the spectrum, panels b, and c of

Fig 2) that markedly overlap the metabolite signals in the $^1$H-NMR spectrum. This overlapping could hinder a correct integration of the signals of the individual metabolites, requiring a deconvolution process for a correct quantification. However, this advantage was not present compared to the UUM method, where the additional filtration process in the centrifugal filters eliminated all interferences with a superior quality spectrum and better line base (see the red boxes in the spectrum, panels a and c of Fig 2). Moreover, the increased additional signal-to-noise ratio of the spectra obtained with the UUM method compared with FM, enhances a correct integration and quantification of the signals.

It is also interesting to focus on the extraction and lysis processes required to obtain samples for NMR-based metabolomic profiling. These workups are often the most laborious and rate-limiting steps in metabolomics because they require accuracy, repeatability and reproducibility, as well as robustness. To date, several rigorous studies recommend the use of the methanol–chloroform–water mixture (FM) to extract the largest number of metabolites with repeatability [62, 63]. However, following FM is associated with many sequential manipulations (i.e., quenching, multiple lysis, centrifugation, solvent elimination, sample drying, etc.) that significantly increase the risk of experimental errors or the introduction of variability. Instead, the methods herein developed (UM and UUM) involve fewer steps (i.e., simultaneous quenching-lysis-extracting cycles, centrifugation and ultrafiltration), which take place in the same container and minimize the risk of sample loss. Both UM and UUM provided higher reproducibility and repeatability of the results compared to FM, as represented in Figs 4–6.

Moreover, the processing time is another crucial variable. Table 3 denotes how FM takes 2-5-fold longer than UUM and takes 4-fold longer than UM; this is translated into higher sample handling costs and fewer samples analyzed per day. The latter is perhaps the most critical variable in clinics where many samples have to be evaluated, and the results need to be obtained in the shortest possible time. In addition, the need to use toxic solvents (methanol and chloroform) and disposable materials increases the cost of FM. While processing a sample by FM, water, methanol, chloroform, buffer, and at least three centrifuge tubes, are required. With UM, only buffer and one centrifugal tube are needed. UUM requires an additional centrifugal filter, which slightly increases the price of this process compared to UM. FM involves using a freeze dryer, an instrument with a relatively high cost compared to the ultrasound probe required for UM and UUM. The metabolites detected from PBMCs can provide valuable information to diagnose and manage diseases given their nature and function in the human body. The immune system involves two fundamental types of responses in which PBMCs perform functions: the humoral response (Humoral Immunity) and the cell-mediated one (Cellular immunity). These responses are related to the activation of T- and B-lymphocytes. Likewise, in immune defense lines (the innate and adaptive immune systems), PBMCs also play a role. In the innate system, in which cells perform an effector function without requiring specific antigen recognition, NK cells perform related functions, whereas dendritic cells (a type of PBMCs) form a critical interface between both innate and adaptive systems. It should be noted that the medical and therapeutic interest in PBMCs, which has led to state-of-the-art developments in the field of stem cells using a fraction of PBMCs from a single donor, lies in the generation of induced pluripotent stem cells (iPSC), which is extremely relevant in personalized medicine. Moreover, there have been more recent developments, which aim to treat human cancers without a compatible donor by using genome editing technology, such as CRISPR / Cas9, to transform T-cells into CAR-T cells [64]. When the aim is to acquire information about changes in the metabolic state of cells, as is the case of PBMCs, metabolomic profiling by NMR is an analysis method that allows the non-targeted characterization of a large number of different metabolites in a single analysis. This metabolomic technique provides data that can be used in combination with other omics data, such as those obtained by

genomics and proteomics. Taken together, these data have been applied to a wide range of *in vitro* models and have helped to better understand the metabolism of a healthy and a diseased individual [65]. Although we know that some studies exist on the metabolomic profiling of blood cells, most of them have been carried out on red blood cells (RBC) and polymorphonuclear cells (PMNs) [66–74], and very limited data about analyzing PBMCs by NMR spectroscopy from patients are available [38, 75].

In future works, this method can be directly applied to perform high-throughput metabolomics analyses in clinical studies, especially for studying diseases in which the immune system plays an important role. In particular, we intend to apply it to research about the human immunodeficiency virus (HIV), to identify biomarkers and metabolic pathways associated with AIDS development.

## Conclusions

This work presents a new processing method of PBMCs for metabolomic profiling by NMR spectroscopy. High quality, robust and reproducible data can be obtained from PBMC samples of 12.5 million cells (half the amount previously reported) by combining high-intensity ultrasound and centrifugal filtration. The resulting ultrasound and ultrafiltration method (UUM) is characterized by minimum sample handling (the whole process can be done in the same vial) and a short processing time (6 h vs. 16 h that the traditional method lasts). In combination with the easy availability of PBMC samples from patients, methods open up new avenues for the application of $^1$H-NMR-based PBMC metabolomics profiling for disease diagnosis and management.

## Supporting information

**S1 Table. Unnormalized integration data.**
(PDF)

**S2 Table. Normalized integration data.**
(PDF)

**S1 Fig. Six Sigma Gage R&R Measure.** Comparative repeatability and reproducibility analysis between FM, UM and UUM, for the normalized concentration of Alanine.
(TIF)

**S2 Fig. Six Sigma Gage R&R Measure for Valine.** Comparative repeatability and reproducibility analysis between FM, UM and UUM, for the normalized concentration of Valine.
(TIF)

**S3 Fig. Six Sigma Gage R&R Measure for Taurine.** Comparative repeatability and reproducibility analysis between FM, UM and UUM, for the normalized concentration of Taurine.
(TIF)

**S4 Fig. Six Sigma Gage R&R Measure for Inosine.** Comparative repeatability and reproducibility analysis between FM, UM and UUM, for the normalized concentration of Inosine.
(TIF)

**S5 Fig. Six Sigma Gage R&R Measure for Hidroxyacetone.** Comparative repeatability and reproducibility analysis between FM, UM and UUM, for the normalized concentration of Hidroxyacetone.
(TIF)

**S6 Fig. Six Sigma Gage R&R Measure for Creatinine.** Comparative repeatability and reproducibility analysis between FM, UM and UUM, for the normalized concentration of Creatinine.
(TIF)

**S7 Fig. Six Sigma Gage R&R Measure for Creatine.** Comparative repeatability and reproducibility analysis between FM, UM and UUM, for the normalized concentration of Creatine.
(TIF)

**S8 Fig. Six Sigma Gage R&R Measure for Choline.** Comparative repeatability and reproducibility analysis between FM, UM and UUM, for the normalized concentration of Choline.
(TIF)

## Author Contributions

**Conceptualization:** León Gabriel Gómez-Archila, Martina Palomino-Schätzlein, Elkin Galeano.

**Data curation:** León Gabriel Gómez-Archila, Martina Palomino-Schätzlein, Elkin Galeano.

**Formal analysis:** León Gabriel Gómez-Archila, Martina Palomino-Schätzlein.

**Funding acquisition:** Martina Palomino-Schätzlein, Wildeman Zapata-Builes, Elkin Galeano.

**Investigation:** León Gabriel Gómez-Archila.

**Methodology:** León Gabriel Gómez-Archila, Martina Palomino-Schätzlein, Elkin Galeano.

**Project administration:** León Gabriel Gómez-Archila, Wildeman Zapata-Builes, Elkin Galeano.

**Resources:** Wildeman Zapata-Builes, Elkin Galeano.

**Supervision:** Elkin Galeano.

**Validation:** Martina Palomino-Schätzlein.

**Visualization:** León Gabriel Gómez-Archila, Martina Palomino-Schätzlein.

**Writing – original draft:** León Gabriel Gómez-Archila, Martina Palomino-Schätzlein, Wildeman Zapata-Builes, Elkin Galeano.

**Writing – review & editing:** León Gabriel Gómez-Archila, Martina Palomino-Schätzlein, Wildeman Zapata-Builes, Elkin Galeano.

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
