## [Decision Letter · Decision Letter 0]

5 Aug 2020

PONE-D-20-14510

Development of an optimized method of processing Peripheral Blood Mononuclear Cells for 1H-Nuclear Magnetic Resonance based metabolomic profiling

PLOS ONE

Dear Dr. Gómez Archila,

Thank you for submitting your manuscript to PLOS ONE. After careful consideration, we feel that it has merit but does not fully meet PLOS ONE’s publication criteria as it currently stands. Therefore, we invite you to submit a revised version of the manuscript that addresses the points raised during the review process.

Specifically, there are a number of issues that must be addressed before the manuscript can be considered for publication.

We look forward to receiving your revised manuscript.

Kind regards,

Oscar Millet

Academic Editor

PLOS ONE

Journal Requirements:

2. Please provide further details on participant numbers, demographics and recruitment. Please also provide the IRB approval number in the ethics statement."

3. We note that this submission includes NMR spectroscopy data. We would recommend that you include the following information in your methods section or as Supporting Information files:

a) The make/source of the NMR instrument used in your study, as well as the magnetic field strength. For each individual experiment, please also list: the nucleus being measured; the sample concentration; the solvent in which the sample is dissolved and if solvent signal suppression was used; the reference standard and the temperature.

b. A list of the chemical shifts for all compounds characterised by NMR spectroscopy, specifying, where relevant: the chemical shift (δ), the multiplicity and the coupling constants (in Hz), for the appropriate nuclei used for assignment.

c. The full integrated NMR spectrum, clearly labelled with the compound name and chemical structure.

We also strongly encourage authors to provide primary NMR data files, in particular for new compounds which have not been characterised in the existing literature. Authors should provide the acquisition data, FID files and processing parameters for each experiment, clearly labelled with the compound name and identifier, as well as a structure file for each provided dataset. See our list of recommended repositories here: https://journals.plos.org/plosone/s/recommended-repositories

4. We noticed you have some minor occurrence of overlapping text with previous publications, which needs to be addressed. In your revision ensure you cite all your sources (including your own works), and quote or rephrase any duplicated text outside the methods section. Further consideration is dependent on these concerns being addressed.

5.

We suggest you thoroughly copyedit your manuscript for language usage, spelling, and grammar. If you do not know anyone who can help you do this, you may wish to consider employing a professional scientific editing service.  

Reviewers' comments:

Reviewer's Responses to Questions

**Comments to the Author**

1. Is the manuscript technically sound, and do the data support the conclusions?

Reviewer #1: No

2. Has the statistical analysis been performed appropriately and rigorously? 

Reviewer #1: No

3. Have the authors made all data underlying the findings in their manuscript fully available?

Reviewer #1: No

4. Is the manuscript presented in an intelligible fashion and written in standard English?

Reviewer #1: No

5. Review Comments to the Author

Reviewer #1: Gómez Archila et al. have evaluated two new methods for PBMCs metabolomic profiling by using NMR spectroscopy: Ultrasound Method (UM) and Ultrasound and Ultrafiltration Method (UUM). They have compared them against the gold standard Folch method in terms of sensitivity, processing time, spectrum quality, metabolite numbers and reproducibility.

The present study aims to apply a previously validated methodology based on high intensity ultrasounds sample processing to obtain PBMC extracted matrixes for NMR metabolomics profiling. The authors argue a higher degree of robustness and reproducibility by using these new methods, and reduced sample handling time compared to the traditional Folch extraction methods.

The sample size is not described, the conclusions are only qualitative, there’s no statistical treatment of results, as there’s not an analytical approach. The methods are interesting, although they should address considerably the study approach.

My major concern -considering the objective and nature of the study- is the absolutely lack of quantitative comparison.

This study should include the analysis of a sample set representing a wider range of biological conditions, to be able to report the analytical performance of the new approach compared against the gold standard. The authors do not report quantitative measurements of robustness, reproducibility, fold changes, sensibility and variability. Neither replicates… as it appears to be 1 sample analysis.

Minor: There are some aspects deserving a better approach:

In the abstract the authors should include methodological aspects.

Line 68- 70 Please, include references and clarify:

“For example in the case of, immunosuppressive diseases (HIV type), cancer and primary immunodeficiencies the availability of PBMCs will be very low.”

Line 70 -74 To Consider Rewriting: I do not understand the construction.

“Therefore, taking into account cross-sectional or longitudinal studies, death, loss of contact, etc., it would be optimal to work with smaller blood and cell samples by changing the isolation method, the processing time (sample handling) delays and/or the cryopreservation technique that can affect the metabolic profile of the PBMCs obtained”.

Methods: Please, include some details in the Human subjects description. It should include some complementary information. I do understand that healthy volunteers accepted to donate blood for research purposes, but how was created the validation cohort? There was some inclusion / exclusion criteria. % hematocrite, fasting, non-fasting, volume extraction, extraction tubes, etc…

General comments: English review is mandatory

6. PLOS authors have the option to publish the peer review history of their article (what does this mean?). If published, this will include your full peer review and any attached files.

Reviewer #1: No

---

## [Author Response · Author response to Decision Letter 0]

27 Nov 2020

Journal Requirements:

We have now revised the PLOS ONE’s style requirement, to ensure that the new version of the manuscripts fulfils these requirements. 

2. Please provide further details on participant numbers, demographics and recruitment. Please also provide the IRB approval number in the ethics statement."

We have now provided the required details in the methods section. 

3. We note that this submission includes NMR spectroscopy data. We would recommend that you include the following information in your methods section or as Supporting Information files:

a) The make/source of the NMR instrument used in your study, as well as the magnetic field strength. For each individual experiment, please also list: the nucleus being measured; the sample concentration; the solvent in which the sample is dissolved and if solvent signal suppression was used; the reference standard and the temperature.

To our knowledge, all this information was already included in the methods section. We grateful if you could give us more detail if there is still any information missing. 

b. A list of the chemical shifts for all compounds characterised by NMR spectroscopy, specifying, where relevant: the chemical shift (δ), the multiplicity and the coupling constants (in Hz), for the appropriate nuclei used for assignment.

We have now included the chemical shift information of each compound in Table 1. As we have only assigned already known compounds, the information about multiplicity and coupling constants can be easily found in databases (e. g. Human Metabolome Database https://hmdb.ca/), wherefore we think there is no need to include it in our work. 

c. The full integrated NMR spectrum, clearly labelled with the compound name and chemical structure.

We also strongly encourage authors to provide primary NMR data files, in particular for new compounds which have not been characterised in the existing literature. Authors should provide the acquisition data, FID files and processing parameters for each experiment, clearly labelled with the compound name and identifier, as well as a structure file for each provided dataset. See our list of recommended repositories here: https://journals.plos.org/plosone/s/recommended-repositories

NMR spectra obtained with the three extraction methods are depicted in Figures 2 and 3. As these spectra are spectra from mixtures and not from pure compounds, it would be a bit confusing to include an integration of the signals in the spectra. However, we have now included a data table with the raw integration values in the supporting information. 

4. We noticed you have some minor occurrence of overlapping text with previous publications, which needs to be addressed. In your revision ensure you cite all your sources (including your own works), and quote or rephrase any duplicated text outside the methods section. Further consideration is dependent on these concerns being addressed.

We have now rephrased the text in the first paragraph of the introduction and the lasts paragraphs of the discussion in order to avoid overlapping with previous publication. We have also added a new reference in the discussion section. 

5.

We suggest you thoroughly copyedit your manuscript for language usage, spelling, and grammar. If you do not know anyone who can help you do this, you may wish to consider employing a professional scientific editing service. 

The language usage in the new version of the manuscript has been revised by the professional service of Helen Warburton, member of the AEPD.

5. Review Comments to the Author

Reviewer #1: Gómez Archila et al. have evaluated two new methods for PBMCs metabolomic profiling by using NMR spectroscopy: Ultrasound Method (UM) and Ultrasound and Ultrafiltration Method (UUM). They have compared them against the gold standard Folch method in terms of sensitivity, processing time, spectrum quality, metabolite numbers and reproducibility.

The present study aims to apply a previously validated methodology based on high intensity ultrasounds sample processing to obtain PBMC extracted matrixes for NMR metabolomics profiling. The authors argue a higher degree of robustness and reproducibility by using these new methods, and reduced sample handling time compared to the traditional Folch extraction methods.

The sample size is not described, the conclusions are only qualitative, there’s no statistical treatment of results, as there’s not an analytical approach. The methods are interesting, although they should address considerably the study approach.

My major concern -considering the objective and nature of the study- is the absolutely lack of quantitative comparison.

This study should include the analysis of a sample set representing a wider range of biological conditions, to be able to report the analytical performance of the new approach compared against the gold standard. The authors do not report quantitative measurements of robustness, reproducibility, fold changes, sensibility and variability. Neither replicates… as it appears to be 1 sample analysis.

We thank the reviewer for the evaluation. We have now completed our work with additional experiments and a more quantitative approach. In our initial study, we analysed three replicate samples from one patient for each method. In the study presented in the new manuscript, we have added an analysis of three more patients, acquire duplicates at two different measuring days. Furthermore, we now have done a quantitative statistical comparison of concentration values of all metabolites, and have calculated repeatability and reproducibility for selected, representative metabolites. We have also calculated the detection and quantification limits of the three methods. All this information has been included in the new results and discussion sections. 

Minor: There are some aspects deserving a better approach:

In the abstract the authors should include methodological aspects.

We have now added more methodological details in the abstract.

Line 68- 70 Please, include references and clarify:

“For example in the case of, immunosuppressive diseases (HIV type), cancer and primary immunodeficiencies the availability of PBMCs will be very low.”

We have now rephrased the sentence and added new references. 

Line 70 -74 To Consider Rewriting: I do not understand the construction.

“Therefore, taking into account cross-sectional or longitudinal studies, death, loss of contact, etc., it would be optimal to work with smaller blood and cell samples by changing the isolation method, the processing time (sample handling) delays and/or the cryopreservation technique that can affect the metabolic profile of the PBMCs obtained”.

We thank the reviewer for this comment, indeed the construction was a bit confusing. We have rephrased the sentence in the new version of the manuscript. 

Methods: Please, include some details in the Human subjects description. It should include some complementary information. I do understand that healthy volunteers accepted to donate blood for research purposes, but how was created the validation cohort? There was some inclusion / exclusion criteria. % hematocrite, fasting, non-fasting, volume extraction, extraction tubes, etc…

We have now provided the required details about the volunteers in the methods section. 

General comments: English review is mandatory

The language usage in the new version of the manuscript has been revised by the professional service of Helen Warburton, member of the AEPD.

---

## [Decision Letter · Decision Letter 1]

29 Dec 2020

PONE-D-20-14510R1

Development of an optimized method of processing Peripheral Blood Mononuclear Cells for 1H-Nuclear Magnetic Resonance based metabolomic profiling

PLOS ONE

Dear Dr. Gómez Archila,

Thank you for submitting your manuscript to PLOS ONE. After careful consideration, we feel that it has merit but does not fully meet PLOS ONE’s publication criteria as it currently stands. Therefore, we invite you to submit a revised version of the manuscript that addresses the points raised during the review process.

There are still a number of issues raised by the reviewer that need to be addressed before we can further proceed with the manuscript.

We look forward to receiving your revised manuscript.

Kind regards,

Oscar Millet

Academic Editor

PLOS ONE

Reviewers' comments:

Reviewer's Responses to Questions

**Comments to the Author**

1. If the authors have adequately addressed your comments raised in a previous round of review and you feel that this manuscript is now acceptable for publication, you may indicate that here to bypass the “Comments to the Author” section, enter your conflict of interest statement in the “Confidential to Editor” section, and submit your "Accept" recommendation.

Reviewer #1: (No Response)

2. Is the manuscript technically sound, and do the data support the conclusions?

Reviewer #1: Yes

3. Has the statistical analysis been performed appropriately and rigorously? 

Reviewer #1: I Don't Know

4. Have the authors made all data underlying the findings in their manuscript fully available?

Reviewer #1: Yes

5. Is the manuscript presented in an intelligible fashion and written in standard English?

Reviewer #1: Yes

6. Review Comments to the Author

Reviewer #1: Gómez Archila et al. have evaluated two new methods for PBMCs metabolomic profiling by using NMR spectroscopy: Ultrasound Method (UM) and Ultrasound and Ultrafiltration Method (UUM). They have compared them against the gold standard Folch method in terms of sensitivity, processing time, spectrum quality, metabolite numbers and reproducibility.

The present study has addressed previous concerns and has really improved validating their methodology for NMR metabolomics profiling. Although some minor questions remained opened:

Line 83: Please, consider clarifying the highlighted term:

For example, with immunosuppressive diseases such as HIV [39], cancer [40] and primary immunodeficiencies, characterized by lower leukocyte production, the availability of PBMCs reduces.

Line 199:

Please indicate the nuclear Overhauser Effect spectroscopy.

Line 218: Please, consider including into the discussion section the limitation associated with integrative methods to obtain metabolite concentration as the one presented in the manuscript; and the potential benefits of new UUM / UM methods on alternative mathematical treatment of the raw spectra based on methods (deconvolution processes). Would this approach differently improve quantification in any of the described experimental methods?

Line 223: The statistical analysis should be carefully reviewed. There are different objectives, as it can be easily seen in the Results, but not in the Data analysis and statistics methods section. I would recommend to emphasize the different objectives (to evaluate the reproducibility, by using the variation coefficient, to evaluate the differences in the number between methods, and also in the area by difference on central tendency, ANOVA, etc.... It looks like it has been evaluated together. So, clarify how do you address the differences between central tendences? And how do you treat the duplicates/triplicates (not as independent / nor paired measurements).

Line 224 Please, consider using the term mean in Table 2 instead of Average.

Line 227: Please review the use of central tendency instead of mean. Mean differences for t-tests, and median differences for Wilcoxon test.

Statistics general comment: There's a problem with this particularly low sample size regarding the power of the test. The authors should state that in the discussion, groups are not large enough to have high power of detecting a meaningful difference say delta between the two means, but artifacts. You should add this limitation for the t test and the normality test.

Line 233: central difference instead of mean (as it depends on the test)

Line 254: From figure it is not easy to identify differences between Ultrasound and Ultrafiltration methods. The two methods are explained or only the UUM? please clarify as there’s a plural form in 256.

7. PLOS authors have the option to publish the peer review history of their article (what does this mean?). If published, this will include your full peer review and any attached files.

Reviewer #1: **Yes: **Dr. Nuria Amigo Grau

---

## [Author Response · Author response to Decision Letter 1]

4 Feb 2021

Reviewers' comments:

Reviewer's Responses to Questions

Comments to the Author

1. If the authors have adequately addressed your comments raised in a previous round of review and you feel that this manuscript is now acceptable for publication, you may indicate that here to bypass the “Comments to the Author” section, enter your conflict of interest statement in the “Confidential to Editor” section, and submit your "Accept" recommendation.

Reviewer #1: (No Response)

2. Is the manuscript technically sound, and do the data support the conclusions?

Reviewer #1: Yes

3. Has the statistical analysis been performed appropriately and rigorously?

Reviewer #1: I Don't Know

4. Have the authors made all data underlying the findings in their manuscript fully available?

Reviewer #1: Yes

5. Is the manuscript presented in an intelligible fashion and written in standard English?

Reviewer #1: Yes

6. Review Comments to the Author

Reviewer #1: Gómez Archila et al. have evaluated two new methods for PBMCs metabolomic profiling by using NMR spectroscopy: Ultrasound Method (UM) and Ultrasound and Ultrafiltration Method (UUM). They have compared them against the gold standard Folch method in terms of sensitivity, processing time, spectrum quality, metabolite numbers and reproducibility.

The present study has addressed previous concerns and has really improved validating their methodology for NMR metabolomics profiling. Although some minor questions remained opened:

Line 83: Please, consider clarifying the highlighted term:

For example, with immunosuppressive diseases such as HIV [39], cancer [40] and primary immunodeficiencies, characterized by lower leukocyte production, the availability of PBMCs reduces.

We have now rephrased the sentence to make it clearer. 

Line 199:

Please indicate the nuclear Overhauser Effect spectroscopy.

We have now indicated the term Nuclear Overhauser Effect Spectroscopy at the recommended place.

Line 218: Please, consider including into the discussion section the limitation associated with integrative methods to obtain metabolite concentration as the one presented in the manuscript; and the potential benefits of new UUM / UM methods on alternative mathematical treatment of the raw spectra based on methods (deconvolution processes). Would this approach differently improve quantification in any of the described experimental methods?

We have now included limitations and improvements of the new UUM/UM methods concerning the integration procedure in the third paragraph of the discussion. 

Line 223: The statistical analysis should be carefully reviewed. There are different objectives, as it can be easily seen in the Results, but not in the Data analysis and statistics methods section. I would recommend to emphasize the different objectives (to evaluate the reproducibility, by using the variation coefficient, to evaluate the differences in the number between methods, and also in the area by difference on central tendency, ANOVA, etc.... It looks like it has been evaluated together. So, clarify how do you address the differences between central tendences? And how do you treat the duplicates/triplicates (not as independent / nor paired measurements).

We have now restructured the data analysis and statistics Materials and Methods section and added more details in order to clarify your questions about data analysis and statistics.

Line 224 Please, consider using the term mean in Table 2 instead of Average.

We have now changed the term “average” for “mean” in Table 2 according to your recommendation.

Line 227: Please review the use of central tendency instead of mean. Mean differences for t-tests, and median differences for Wilcoxon test.

We have now changed the term “measures of central tendency” for “means”, specifying “mean difference” for t-test and “median difference” for Wilcoxon test.

Statistics general comment: There's a problem with this particularly low sample size regarding the power of the test. The authors should state that in the discussion, groups are not large enough to have high power of detecting a meaningful difference say delta between the two means, but artifacts. You should add this limitation for the t test and the normality test.

We have now included the limitations mentioned by the referee in the first paragraph of the discussion. 

Line 233: central difference instead of mean (as it depends on the test)

We have now changed the term “central (measures of central tendency)” instead of “mean”.

Line 254: From figure it is not easy to identify differences between Ultrasound and Ultrafiltration methods. The two methods are explained or only the UUM? please clarify as there’s a plural form in 256.

We have now changed the figure caption to make clear that only the UUM method is represented on the figure. 

7. PLOS authors have the option to publish the peer review history of their article (what does this mean?). If published, this will include your full peer review and any attached files.

Do you want your identity to be public for this peer review? For information about this choice, including consent withdrawal, please see our Privacy Policy.

Reviewer #1: Yes: Dr. Nuria Amigo Grau

---

## [Editor Report · Decision Letter 2]

11 Feb 2021

Development of an optimized method of processing Peripheral Blood Mononuclear Cells for 1H-Nuclear Magnetic Resonance based metabolomic profiling

PONE-D-20-14510R2

Dear Dr. Gómez Archila,

We’re pleased to inform you that your manuscript has been judged scientifically suitable for publication and will be formally accepted for publication once it meets all outstanding technical requirements.

Kind regards,

Oscar Millet

Academic Editor

PLOS ONE
---

## [Editor Report · Acceptance letter]

15 Feb 2021

PONE-D-20-14510R2 

Development of an optimized method for processing Peripheral Blood Mononuclear Cells for 1H-Nuclear Magnetic Resonance-based metabolomic profiling 

Dear Dr. Gómez-Archila:

I'm pleased to inform you that your manuscript has been deemed suitable for publication in PLOS ONE. Congratulations! Your manuscript is now with our production department. 

Kind regards, 

on behalf of

Dr. Oscar Millet 

Academic Editor

PLOS ONE